# The Effectiveness of Educational Training or Multicomponent Programs to Prevent the Use of Physical Restraints in Nursing Home Settings: A Systematic Review and Meta-Analysis of Experimental Studies

**DOI:** 10.3390/ijerph17186738

**Published:** 2020-09-16

**Authors:** Anna Brugnolli, Federica Canzan, Luigina Mortari, Luisa Saiani, Elisa Ambrosi, Martina Debiasi

**Affiliations:** 1Centre of Higher Education for Health Sciences, 38122 Trento, Italy; anna.brugnolli@apss.tn.it; 2Department of Diagnostics and Public Health, University of Verona, 37134 Verona, Italy; federica.canzan@univr.it (F.C.); luisa.saiani@univr.it (L.S.); elisa.ambrosi_01@univr.it (E.A.); 3Department of Human Sciences, University of Verona, 37134 Verona, Italy; luigina.mortari@univr.it

**Keywords:** nursing homes, older people, physical restraint, systematic review, meta-analysis

## Abstract

This review assesses the effectiveness of interventions to reduce physical restraint (PR) use in older people living in nursing homes or residential care facilities. A systematic search of studies published in four electronic databases (MEDLINE, CINHAL, PsycINFO, Cochrane Central Register of Controlled Trials). The review included individual and cluster randomized controlled trials that compared educational training and multicomponent programs to avoid PR use. Risk bias of randomized controlled trials (RCTs) was assessed according to the Cochrane Handbook for Systematic Reviews of Interventions. This review includes 16 studies in a qualitative synthesis that met the inclusion criteria, nine of them offered a multicomponent program and seven offered only educational training. The results of the 12 studies included in the meta-analysis showed a significant trend in favor of intervention over time and intensity of PR use tends to decrease. The review indicates that educational programs and other supplementary interventions should be effective, but the heterogeneous operative definition of physical restraints can make difficult data generalization.

## 1. Literature Review

Physical restraints (PR) are still commonly used worldwide in nursing homes and in residential care facilities, even if previous epidemiological studies showed wide differences between and within countries [1,2]. The main predictors of physical restraint use are patient’s impaired mobility, impaired cognitive status, and risk of falls; organizational policies can also determine their use. Health care professionals declared, as reported in different qualitative studies, that they use PR not only for preventing falls and injuries but also for controlling dangerous behaviors and for preventing interference with medical devices, such as urinary catheter and nasogastric tubes [3,4,5,6,7,8]. However, current evidence does not support PR effectiveness in reducing and preventing falls and fall-related injuries and questions their safety [3,9,10]. In the last years, a restraint-free nursing care environment has been recommended as a standard of care; therefore, policies and regulations aiming at reducing PR use have been implemented in many countries [11]. This topic is articulated and full of conceptual and operative details. An international consensus statement defines physical restraint as “any device, material or equipment attached to or near a person’s body and which cannot be controlled or easily removed by the person, and which deliberately prevents or is deliberately intended to prevent a person’s free body movement to a position of choice and/or a person’s normal access to their body” [12]. The most commonly methods used are bilateral bed rails, limb or trunk belts, fixed tables on a chair or chairs which prevent patients getting up on their own, and containment sheets and pajamas [3]. A European study conducted on nursing home residents suffering from dementia revealed a prevalence of physical restraints of 31.4%, with a country variation ranging from 6.1% to 83.2% [13]. However, in USA the prevalence ranged from 6.9% to 36.8% [7]; it was estimated at 31.4% in Canada and at 20.2% in China [1]. The differences in the observed occurrence of PR use are influenced by a variety of factors, such as population studied, clinical context, definition of PR and study design [14]. In recent years, restraint-free care has focused on what interventions or strategies can be employed to promote the reduction of physical restraint [15,16,17]. The use of PR is not without risks; adverse effects can be observed ranging from bodily injuries, decreased mobility, and reduced physiological well-being and up to death [3,18,19,20]. Morover PR use raises ethical issues affecting human rights and indeed several authors consider this practice as an approach that violates the freedom and dignity of restrained residents [15,21,22,23]. The persistent use of physical restraints in nursing homes requires effective interventions for educating nursing staff as well as all persons involved in healthcare, for example residents, relatives, nursing experts and nursing homes directors. Several and articulate interventions aimed to address nursing care and institution’s organizational culture about physical restraint use have been studied. However, previous systematic reviews have shown inconclusive evidence about the effectiveness of educational or multicomponent interventions for preventing and reducing the use of physical restraints in long term geriatric care [2,24]. Therefore, it is important to analyze the effectiveness of specific interventions, such as educational programs and consultation or guidance by an expert nurse, to reduce the use of PR in nursing home settings. The challenge of clinicians and researchers remains to find the ideal mix of interventions to avoid the use of physical restraints from clinical practice.

## 2. Materials and Methods

### 2.1. Aims

The aims of this systematic review are to assess the effectiveness of an educational training or multicomponent program: (1) in preventing the use of PR in nursing home, and (2) in changing (increasing or decreasing) the rate of patient falls and fall-related injuries.

### 2.2. Design

A systematic review with meta-analysis was conducted based on the Cochrane Handbook for Systematic Review for Intervention [25] and performed following the Guidelines for reporting Systematic Reviews and Meta-Analyses (PRISMA checklist criteria, see Appendix A). This review protocol has been registered on the PROSPERO International prospective register of systematic reviews (PROSPERO 2018: CRD42019127963).

### 2.3. Search Methods

Studies published from 1996 to September 2019 were searched using four electronic databases (MEDLINE, CINAHL, the Cochrane Library, and PsycINFO). Other databases, such as www.ClinicalTrials.gov, EU Clinical Trial register, PROSPERO, and Cochrane Library, were searched for unpublished or ongoing studies, as well as reference lists of other studies, all of which were included in this study, were checked. The search strategy used is reported in Appendix B. The following word and MeSH terms were used: Restraint, Physical [Mesh]; Nursing Homes [Mesh] Residential Facilities [Mesh], “bed rail”, “side rail”, “coercion”, and “education”. Inclusion criteria were (a) published and unpublished studies that used cluster and individually randomized controlled trials and quasi-experimental research designs, (b) studies that compare educational training and multicomponent programs to avoid the use of PR on residents in nursing home settings were included in this study, (c) studies written in Italian and English language. The primary outcome was the status of PR as defined by Bleijlevens et al. [12] measured by number of patients who were physically restrained at the studies endpoint, assessed either by direct observation or from clinical documentation, or calculating restraint prevalence (number of restraints/number of residents × 100). Moreover, falls rate and falls- related-injuries) were assessed as secondary outcomes. Studies excluded were those that referred to psychiatric and critical populations in acute ward or home care.

### 2.4. Search Outcome

The electronic and manual reference list search revealed 418 publications. A total of 122 references that could meet the inclusion criteria were screened, and 222 publications were excluded; another 82 were excluded for outcomes other than “physical restraint rate”, incomplete data and being written in other languages (German and French). The full texts of the remaining 40 articles were reviewed, and of these, 16 were included in qualitative synthesis and 12 in meta-analysis (Figure 1. PRISMA Flow Diagram).

### 2.5. Quality Appraisal

Two reviewers (Anna Brugnolli and Martina Debiasi) independently screened the title and abstracts that met the inclusion criteria, and then screened the full text of potentially included studies. The citations in full-text articles marked as included were retrieved and those citations that the reviewers were unsure of were excluded. Disagreements and discrepancies were resolved by consensus, and when necessary, by consultation and discussion within the reviewer teams during all stages of the review process (Federica Canzan, Luigina Mortari, Luisa Saiani and Elisa Ambrosi). Risk bias of randomized controlled trials (RCTs) was assessed according to the Cochrane Handbook for Systematic Reviews of Interventions [24]. For RCTs, we considered random sequence generation, allocation concealment, blinding of participants and personnel, blinding of outcome assessment, incomplete outcome data, selective outcome reporting, and other bias. For cluster randomized trials we evaluated recruitment bias, baseline imbalance, loss of clusters, incorrect analysis, and comparability with individually RCTs.

### 2.6. Data Abstractions

For each included study data were extracted by two independent reviewers using a standardized form and checked for accuracy by a third reviewer. The results were discussed within the reviewers’ team. A descriptive summary of the included studies was created by drawing on two tables of evidence: a general Characteristics table that includes details about the study type, interventions, characteristics of the population, type of intervention and primary and secondary outcome measures with a parallel definition and measurement of restraint status in the studies included (Appendix A), and a Results table that classified the results for each outcomes.

### 2.7. Synthesis and Supplementary Statistical Analysis

We conducted a meta-analysis using random or fixed effect model where possible, a narrative synthesis was conducted where there was insufficient data. We performed analysis at the level of cluster RCTs. For one study further detail have been requested about the data and effect size about PR used [26]. To assess statistical heterogeneity, we calculated the I^2^ using Review Manager 5.3 (RevMan, Copenhagen, Denmark); the primary analysis used a random-effects model (risk ratio, RR), which had the highest generalizability in our empirical examination of summary effect measures for meta-analyses [24]. If the heterogeneity with random-effect model was I^2^ < 50% we used fixed model to estimate the intervention effects, in contrast, if I^2^ > 50% we used random-effect model.

## 3. Results

### 3.1. Study Characteristics

Sixteen studies met the inclusion criteria as described in Table 1. Eleven were cluster RCTs [9,15,26,27,28,29,30,31,32,33,34]; one individual RCT [35]; three quasi-experimental design studies [36,37,38]; and one pre-posttest [39]. Twelve studies were conducted in nursing homes [15,28,30,31,32,33,34,35,36,37,38,39]; four of these had psychogeriatric units/wards dedicated to residents with dementia and behavioral disorders [29,30,31,37]. One study had a unit for people with dementia [9] and one unit in a care home for people with dementia was included because of its close similarities to a residential care facility [26].

### 3.2. Risk of Bias Assessment of the Included Studies

The quality of RCTs was moderate, while it was low for quasi-experimental studies. Of the 16 included studies, nine had adequate sequence generation. Only three reported allocation concealment [15,27,32], blinding of participants and personnel were not possible in any included studies. Ten studies had blinding of outcome assessment [15,26,27,28,30,31,33,34,37,38].

Two of the included studies were assessed as having addressed incomplete data reporting (high risk of bias) [26,34] and lacked an explanation of the choices that underlined the recruitment of the participants [33,34]. Four studies were high risk for baseline imbalance [26,28,32,34], and in one study this issue was not clear [9]. In half of the studies, the analyses were correctly performed, although in some studies, the modalities were not described in detail [28,30,31,33,34]. Methodological quality assessment of the meta-analysis included studies is reported in Figure 2 and Figure 3, and Appendix C.

### 3.3. Types of Interventions

The characteristics of the intervention group are described in Table 1. Six studies offered only educational training [9,26,33,34,35,39]; and multicomponent programs were performed in ten studies [15,27,28,29,30,31,32,36,37,38]. There were seminars focusing on different topics and based on guideline or best practice followed by guidance or consultation [9,15,26,27]. Educational program changed in time from a minimum of six hours training course to a maximum of 6 months guidance and covering different themes about PR avoiding strategies. The arguments covered by the educational programs addressed the following topics:-Information on dementia, aggression, and challenging behavior (delirium), falls and fall prevention, care of people with dementia, complications in dementia, decision-making processes and alternatives [8,33];-Strategies for analyzing and managing aggression or challenging behaviors [9,28,30,32];-Information about legal implications, adverse effects, experience of feelings of being restrained [9,27,28,30,32];-Alternative strategies to the use of PR and decision-making processes [9,26,27,30,33,36];-Falls and fall prevention [9,28,32]; and-Overview of the current evidence about PR and summary of the guideline recommendations [27].

The multicomponent interventions included educational training and guide or consultation by a nurse specialist at the registered nurse level [30,31,36,37,38]; a master’s—prepared gerontological nurse such as an opinion leader [28]; an Advanced Practice nurse [39]; or a trained nurse with specific education (at least a Bachelor’s in Nursing) [27]. The consultations were structured as a monthly session supervision (from one hour to 12 h) or on demand. The multicomponent intervention in several studies provided for dissemination of the guideline’s content in clinical practice, availability of alternative interventions and/or introduction of at least a restraint policy [15,27,36,37,38].

### 3.4. Outcome Measures

Eight of the sixteen included studies had as the primary outcome the use of any PR [9,15,26,28,29,30,31,32,33,34,35]. Follow-up ranged from one to 24 months. Most of the studies provided a conceptual definition of PR (Appendix A): fourteen studies used a comparable definition of PR even if incomplete; three of these studies did not define the devices of restraint [9,28,32] and other two reported only methods [29,39]. Two studies have not reported any definition [33,35]. Eight studies explored falls and injuries related falls; six studies evaluated the impact of a change in prescription of antipsychotic drugs [9,26,34,37,38]. This review analyzed the prevalence of restraint, but it would be relevant to break it down into time of permanence, duration (continuous/discontinuous) and type of restraints used.

### 3.5. Effect of Interventions

The overall results of the qualitative synthesis (16 studies), reported in Table 2, showed a significant trend in favor of interventions over time and intensity of PR use, except for Testad et al. (2016) [26], which found an increase in the use of containments over time in the intervention group and a tendency for a greater reduction in the control group. In contrast, in some RCTs [30,31], both groups, control and intervention, showed an increase in restraint intensity and multiple restraints over time. 12 RCTs of 16 studies were included in the meta-analysis and including 11 cluster-RCTs and 1 individual RCT [35]. The effects of interventions on the primary outcome were presented according to the type of intervention. The overall effect of the educational program (at study endpoint) in reducing PR use were analyzed a total of 1.186 patients (596 intervention and 590 control group) [9,26,28,33,34].

The combined estimated risk ratio (RR) of use of PR with an educational program was statistically significant (Figure 4a: 0.56, 95% CI 0.45–0.69). There was moderate overall heterogeneity (I^2^ = 40%). At medium term (6–8 months), RR of use of physical restraint is RR 0.74 (95% CI 0.39–1.43), and long term (12–24 months), RR 0.67 (95% CI 0.26–1.75). In both subgroups a substantial heterogeneity emerged (Figure 4b).

The overall effect of the multicomponent program (at endpoint studies) in reducing PR use were analyzed in a total of 16.937 patients (8.002 intervention vs. 8.935 control) [15,27,28,29,30,31,32]. The combined estimated RR of use of PR with a multicomponent program was statistically significant (Figure 5a: RR 0.83, 95% CI 0.73–0.94). There was an effect and statistically significant overall high heterogeneity (I^2^ = 80%). The results at short-term (1–4 months) and medium-term (6–8 months) were slightly statistically significant (Figure 5b). At short-term the RR of use of physical restraint is RR 0.86 (95% CI 0.73–1.02), and medium-term (RR 0.82, 95% CI 0.69–0.98). In both subgroups a substantial heterogeneity emerged (I^2^ = 75% and 84%).

### 3.6. Secondary Outcome Results

The results of secondary outcomes are reported in Table 2. The introduction of an educational or multicomponent intervention does not lead to a statistically significant increase in the rate of falls and fall-related injuries [32,33,36,37,38].

## 4. Discussion

This meta-analysis summarizes the evidence regarding the effectiveness of educational programs and multicomponent interventions on the use of mechanical restraints without an increase in falls, behavioral symptoms, or medication. The studies included in this review were adequate and had from low to moderate risk of bias; many of them demonstrated differences regarding the type of interventions and also most of the studies did not have an unvarying definition of physical restraint and methodologies for data collection. A previous Cochrane review underlines that there were insufficient evidence supporting the effectiveness of educational interventions for preventing or reducing the use of PR in geriatric long-term care [2]. In this systematic review emerged the effectiveness of the educational interventions at the endpoint study, and of the multicomponent program all the follow-up time (from 6 to 24 months), in reducing or preventing the use of PR in nursing home settings. More recent evidence of successful reduction efforts in use of restraints is evident from several countries, which used a multicomponent approach [15,27]. This approach demonstrated a reduction in physical restraint use within nursing home after 6- and 12-months post implementing both a detailed guideline and theory-based multicomponent interventions. Multicomponent educational interventions are designed to change the organizational culture towards a least-restraint policy. It is believed that leadership could essentially contribute to restraint-free nursing care by sensitizing nurses and creating optimal working conditions; nursing home staff profile and competencies are appropriate to meet the increasingly complex needs of residents with dementia [40,41]. Knowing that decision-making is mainly based on individual’s experiences and often is ambiguous, the development of evidence-based guidelines to support decision-making regarding the (non)-use of physical restraints is highly recommended [40]. The decision-making process concerning the application of physical restraints has always to respect the resident’s situation as well as nurse- and organization-related factors. The core educational component addresses the attitudes of nurses, physicians and other healthcare professionals who are involved in making decisions and conceiving policies about PR use. Additional components might target the care environment (e.g., environmental changes, adjustment of staff-patient ratios or staff skill mix, involvement of family members, advocacy), or the organizational culture (e.g., attitudes of the opinion leaders or the management), and may support the implementation of change in other ways (e.g., by providing supervision or guidance for healthcare professionals). Protection is a common reason for the use of PR, and the objective is to reduce the use of this method, while employing educational programs and consultation as strategies. In different countries these includes legislation and attention to creating or revising guidelines for nurses and institutions. In addition, it is important to pay attention to the barriers within long-term contexts that do not allow the reduction of PR; educational support is only a starting point and other interventions are needed. This is sure enough, as the results of the analyzed studies support the implementation of educational programs, even alone or with an expert in advanced clinical nursing capable to give consultation and to endorse changes. It is also important to underline that the effectiveness could be decrease in both approaches and is important a re-training to keep knowledge alive.

## 5. Study Limitations

Several limitations of this review have to be considered. The results have limited validity, transferability and generalizability due to the wide range of definitions of physical restraints, the variability of elements that performed a multicomponent program (guidance, consultation, policy or guideline) and/or different measurements in outcomes. Furthermore, the variability of the sample, in terms of baseline characteristics, high turnover, mortality and transfer drop-out or other factors led to an imbalance in the data when comparing the baseline with the data at the follow-up. Finally, there was a lack of data on the proportion of patients with dementia in most of the included studies.

## 6. Relevance to Clinical Practice

The results of this review will be highly relevant for clinical practice and help nurses’ decision-making. Information about effective interventions for preventing and reducing the use of PR in nursing homes may promote care with less or even without PR use which might increase the quality of care of older people. Nursing homes have an important role in the care of older people; the trajectory leading to use of restraint is complex, and the context and nursing factors could affect nurses’ decision-making [42]; further research is thus needed exploring how nurses can be empowered to deal more effectively with this important care issue. Future research is necessary to investigate the feelings and attitudes of healthcare professionals about PR and explore their knowledge of alternatives to PR use. Basic and continuous staff training is a key element in that it allows you to increase knowledge, change attitudes in favor of reducing restraint. This training should include legal, ethical and clinical aspects, with reference to possible alternative ways to avoid restraint in vulnerable people. However, based on the results, training represents a necessary but not enough element, on its own, to create a restraint free culture. In the future, multicomponent programs involving the whole team are needed; and create a multidimensional approach according to professionalism to provide the best patient assistance. Nursing home had to stimulate changes in policy and organizational procedure to reduce or prevent the use of physical restraints. In the included studies definitions of PR are inconclusive and unclear for example, bedside-rails were considered PR in some studies but not in others. Similarly, the educational programs are not always the same and have different elements and components and for these reasons, difficulties arise in comparing the results of the various studies and therefore results are not easy generalizable.

## 7. Conclusions

The results of this review underline that educational training and multicomponent programs could be effective in reducing the use of PR in nursing home settings. At the least, additional studies implementing an educational program alone or with consultation or guidance might provide further evidence of the effectiveness of these approaches on the reduction in the use of PR. The number of older patients with temporary or permanent cognitive impairment in general hospital settings will increase due to demographic change and medical progress. These patients have a higher risk of being restrained [43,44], and PR use may be associated with negative effects that may hamper recovery and rehabilitation. In contrast to geriatric long-term care settings, there is no high-quality systematic review about the effects of interventions intended to prevent or reduce PR use in older people in nursing home settings.

## Figures and Tables

**Figure 1 ijerph-17-06738-f001:**
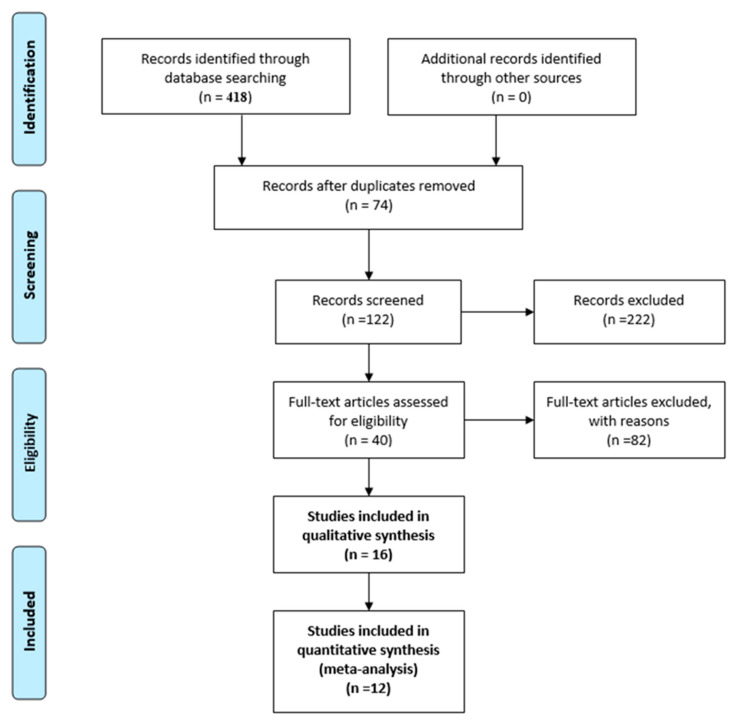
PRISMA Flow Diagram.

**Figure 2 ijerph-17-06738-f002:**
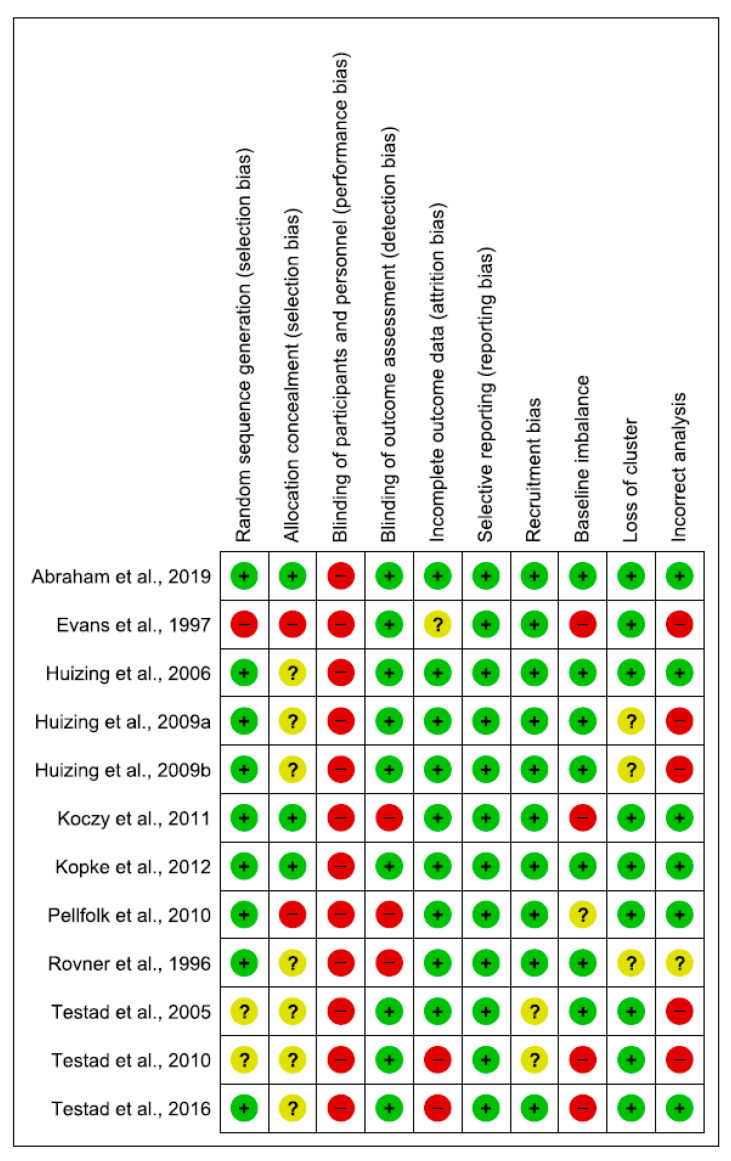
Risk of bias summary.

**Figure 3 ijerph-17-06738-f003:**
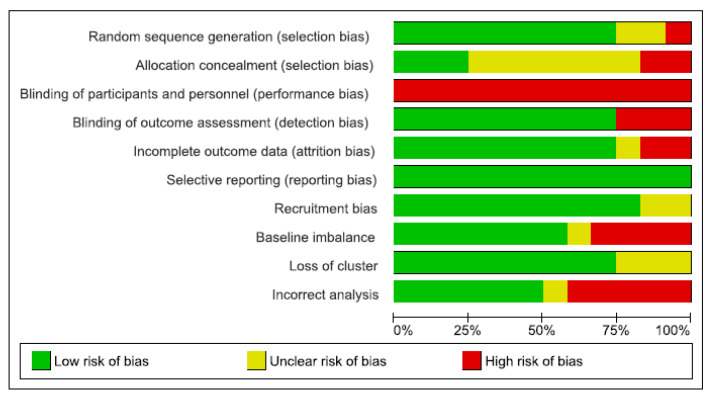
Risk of bias graph: review authors’ judgements about each risk of bias item presented as percentages across all included studies.

**Figure 4 ijerph-17-06738-f004:**
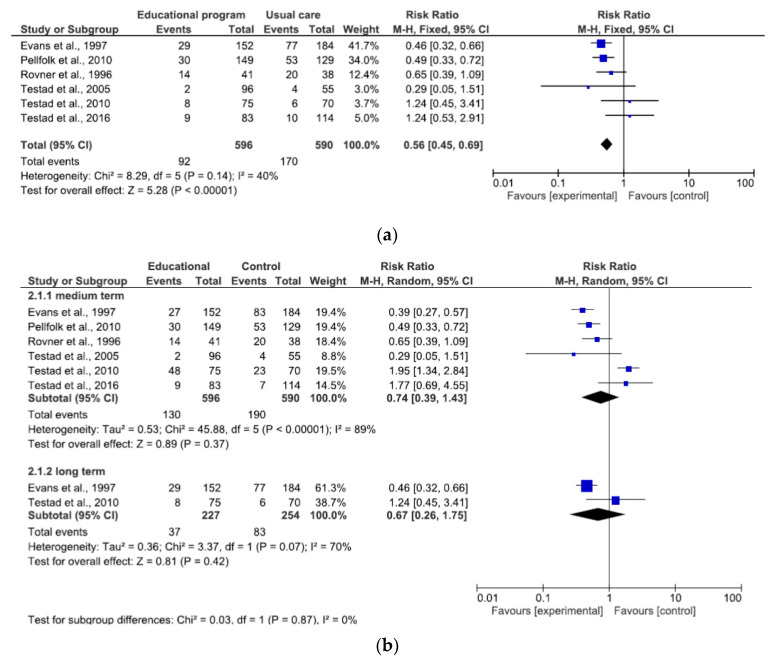
(**a**) Effect of educational program at endpoint study on reducing PR use; (**b**) Effect of educational program at different follow-up on reducing PR use.

**Figure 5 ijerph-17-06738-f005:**
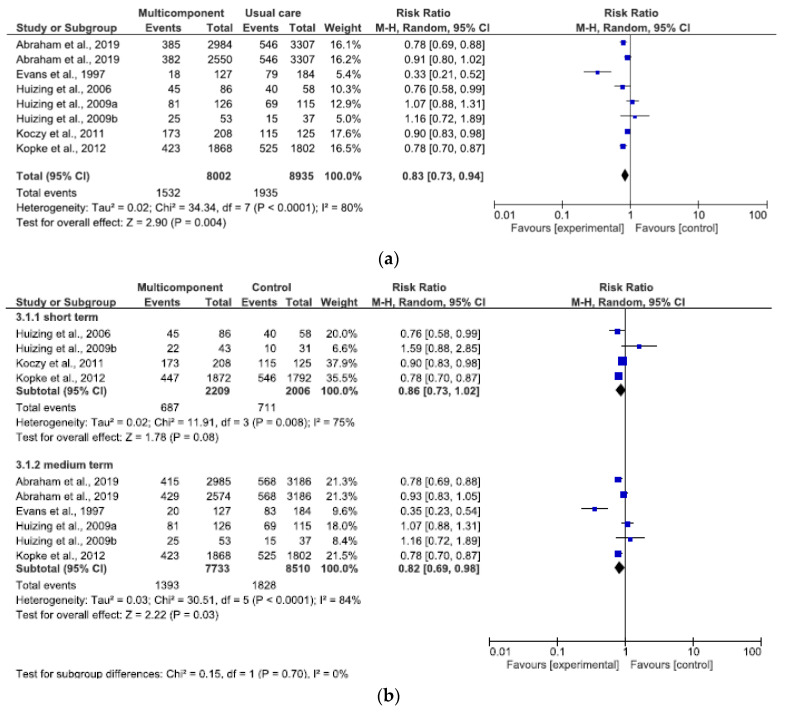
(**a**) Effect of a multicomponent program at endpoint study on reducing PR use; (**b**) Effect of a multicomponent program at different follow-up on reducing PR use.

**Table 1 ijerph-17-06738-t001:** Characteristics of included studies.

Authors (Year) Study Design [Ref]	Country Setting	Sample	Interventions				Control	Follow-Up Period	Primary and Secondary Outcome
			Education Training	Consultation (APN) and Guidance	Change Policy Implementation Guideline	Availability Alternative Intervention			
**Abraham et al. (2019) [27]** Pragmatic RCT	Germany 120 NH	**N = 12,245 residents** IG 1: 4126 IG 2: 3547 CG: 4572	✓		✓	✓	Usual care	6–12 months	Physical restraint use Falls Fall -related fractures Quality of life
**Capezuti et al. (2002) [39]** Pre- and posttest design	USA 4NH	**N = 251 residents (side rail use)** IG: N = 130 Discontinued Side Rail Use IG: N = 121 Continued Side Rail Use		✓			Usual care	1–12 months	Side rail use Incidence of falls
**Evans et al. (1997) [28]** Cluster randomized trial	USA 3 NH	**N = 643 residents** **N = 463 complete data** IGa: 152 IGb: 127 CG: 184	✓	✓			Usual care	6–9–12 months	Physical restraint use Restraint intensity Fall rate
**Gulpers et al. (2011) [37]** Quasi-experimental longitudinal design	Netherlands 26 psycho-geriatric NH	**N = 420 residents** **N = 405 complete data** IG: 250 CG:155	✓	✓	✓	✓	Usual care	4–8 months	Belt restraint use Other types of physical restraint use Use of Psychoactive drugs f Falls Fall-related injuries
**Gulpers et al. (2012) [38]** Quasi-experimental longitudinal study	Netherlands 13 NH	**N = 104 newly admitted residents** **N = 82 complete data base line** IG: t0 43–t1 29 CG: t0 39–t1 20	✓	✓	✓	✓	Usual Care	4–8 months	Belt restraints use e Other types of physical restrains use Use of psychoactive medication Falls Fall-related injuries
**Gulpers et al. (2013) [36]** Quasi-experimental longitudinal study	Netherlands 13 NH	**N = 225 panel group** IG: N = 134 CG: N = 91	✓	✓	✓	✓	Usual care	24 months	belt restraints use Use of at least one physical restraint device
**Huizing et al. (2006) [29]** Cluster randomized trial	Netherlands 5 psycho-geriatric NH	**N = 145 residents with dementia** **N = 126 complete data** IG: t0 83–t1 86 CG: t0 62–t1 58	✓	✓			Usual care	1 month	Physical restraint use Restraint intensity Restraint type
**Huizing et al. (2009a) [30]** Cluster randomized trial	Dutch 7 psycho-geriatric NH	**N = 432 psycho geriatric residents** **N = 241 complete data** IG: N = 125 CG: N = 115	✓	✓			Usual Care	1–4–8 months	Physical restraint status, intensity Multiple restraint use
**Huizing et al. (2009b) [31]** Cluster randomized trial	Dutch 14 psycho-geriatric NH	**N = 138 newly admitted psychogeriatric residents** N = 90 complete data base line IG: 53 CG: 37	✓	✓			Usual Care	1–4–8 months	Physical restraint status, intensity Multiple restraint use
**Koczy et al. (2011) [32]** Cluster randomized trial	Germany 45 NH	**N = 430 restrained residents** N = 333 complete data IG: N = 208 CG: N = 125	✓	✓		✓	Usual care	3 months	**Cessation of physical restraints (100%)**–bed rails were not included Falls Use of psychoactive medication
**Köpke et el. (2012) [15]** Cluster randomized trial	Germany 18 NH	**N = 4449 residents** IG: N = 2283 CG: N = 2166	✓		✓		Standard information	3–6 months	**Physical restraint use (6 month)** Physical restraint use (3 month) Falls Fall-related fractures Use of antipsychotropic therapy
**Pellfolk et al. (2010) [9]** Cluster-randomized controlled trial	Sweden 40 Units for people dementia	**N = 355 residents** N = 350 residents complete data IG: Residents, N = 192 CG: Residents, N = 163	✓				Usual care	6 months	**Physical restraint use** Falls (1 months) Use of antipsychotic therapy (Benzodiazepines–Narcoleptics)
**Rovner et al. (1996) [35]** Randomized controlled Trial (RCT)	Baltimore (USA) A 250-bed community NH	**N = 89 residents randomized** **N = 81 complete data** (91%) IG: N = 42 CG: N = 39	✓				Usual care	6 months	**Behavioral disorders** Use of antipsychotic drugs Physical restraint use Cognition and level of nursing care
**Testad** **(2005) [33]** single-blind cluster Randomised controlled trial	Norway 4NH	**N = 151 residents** IG: N = 55 residents All complete data CG: N = 96 residents N = 87 complete data	✓				Usual care	7 months	**Physical restraint use** Agitation
**Testad et al. (2010) [34]** Single blind cluster randomized controlled trial	Norway 4NH	**N = 211 residents** IG: N = 113 residents N = 76–44 complete data CG: N = 98 residents N = 46 complete data	✓				Usual care	6–12 months	**Structural restraint** Interactional restraint (treatment and care giving activity such as force and pressure) Agitation Use of antipsychotic drugs
**Testad et al. (2016) [26]** Single-blind cluster randomized controlled trial	Norway 24 care homes (citated by authors as NH)	**N = 274 residents** IG: N = 118 residents with dementia N = 85 complete data CG: N = 156 residents N = 116 complete data	✓				Usual care	7 months	**Physical restraint use** Agitation Use of antipsychotic drugs

N = number of residents recruited at baseline; Complete data = residents at study endpoint; NH = nursing home; IG = intervention group; CG = control group.

**Table 2 ijerph-17-06738-t002:** Results of included studies.

Author (Year) [Ref]	MA *	Results on Physical Restraint Use	Results on Fall Rate–Fall-Related Injuries
**Abraham et al. (2019) [27]** A pragmatic cluster of randomized controlled trial	*Y*	Change in any physical restraint prevalence from baseline to follow-up CG −1.2; 95% CI −0.04 to 0.11; *p* = 0.294 IG 1 update version: −2.8; 95% CI −5.5 to −0.01; *p* = 0.042 IG 2 concise version: −3.9; 95% CI −6.8 to −1.0; *p* = 0.009	≥1 Fall 12 months OR (95% CI) IG1 vs. CG: 1.17 (0.89–1.53) OR (95% CI) IG2 vs. CG: 1.03 (0.79–1.35) ≥1 Fall- related fractures OR (95% CI) IG1 vs. CG: 1.31 (0.87–1.97) OR (95% CI) IG2 vs. CG: 1.11 (0.73–1.71)
**Capezuti et al. (2002) [39]** Pre-Post test design	*N*	Side rail use immediately post (1 month) e 12 months Statistically significant effects of time and site, indicating a change over time Only one NH Site 3 showed a statistically significant decrease in the rate of restrictive side rail use over time (*p* = 0.01)	Fall rate 12 months reduced discontinue restrictive side rail group −0.053; 95% CI (−0.083 to −0.024) *p*-value < 0.001 continued restrictive side rail group −0,013; 95% CI (−0.056 to 0.030) *p*-value = 0.17
**Evans et al. (1997) [28]** Cluster RCT	*Y*	Prevalence restraint use (Individual as units of analysis) 6 month CG:45% (83/184); IG: RE: 18% (27/152); REC: 16% (20/127) 9 month CG: 42% (77/184); IG: RE: 16% (24/152); REC: 12% (15/127) 12 month CG: 43% (79/184);IG: RE: 19% (29/152);REC: 14% (18/127) Nursing home as units of analysis 6 month CG:40%; IG: RE: 19%; REC: 18% 9 month CG: 40%; IG: RE: 17%; REC: 14% 12 month CG: 42%; IG: RE: 19%; REC: 16%	Fall rate 3 months GC vs. RE or REC (64.7% vs. 41.5% or 42.5%) *p* < 0.001 6 months GC vs. RE or REC (53.3% vs. 32.2% or 37.8%) *p*-value < 0.001
**Gulpers et al. (2011) [37]** Quasi Experimental	*N*	At least one physical restraint device 4 months CG 64%; IG 54%; *p*-value 0.06 8 months CG 69%; IG 54%; *p*-value 0.003	Falls 4 months GC 14%; GI 20%; *p*-value 0.10 8 months GC 16%; GI 16%; *p*-value 0.98 Fall-related injuries 4 months GC 8%; GI10%; *p*-value 0.44 8 months GC 11%; GI 10%; *p*-value 0.66
**Gulpers et al. (2012) [38]** Quasi Experimental	*N*	At least one physical restraint device 4 months CG 31%; IG 30%; *p*-value 1.00 8 months CG 36%; IG 21%; *p*-value 0.15	Falls 4 months GC 40%; GI 38%; *p*-value 1.00 8 months GC 30%; GI 21%; *p*-value 0.51 Fall-related injuries 4 months GC 10%; GI 24%; *p*-value 0.28 8 months GC 10%; GI 14%; *p*-value 1.00
**Gulpers et al. (2013) [36]** Quasi experimental	*N*	At least one physical restraint 24 months IG: 80/134 (60%); CG: 68/91 (75%) OR 0.50, 95% CI 0.28 to 0.90, *p*-value = 0.020	
**Huizing et al. (2006) [29]** Cluster RCT	*Y*	Restraint use (prevalence) CG 40/58; IG: 45/86 OR 0.50, 95% CI 0.24 to 0.99 *p*-value = 0.048 Restraint intensity over time CG t0 56% t1 70%; IG t0 54%–t1 56% *p*-value > 0.05	
**Huizing et al. (2009a) [30]** cluster RCT	*Y*	Change in restrain status CG 69/115 (60%); IG: 81/126(64%) IG change t0 54% vs. t3 (8 months) 64% (*p* = 0.02)—at post-test 2 there were no differences CG: t0 49% vs. t2 (4 months) 57% (*p* = 0.02); t3 60% (*p* = 0.007)	
**Huizing et al.(2009b) [31]** cluster RCT	*Y*	Not restrained vs. restraint 1 month CG: 70% (14/20)–IG: 61.8% (21/34) vs. CG 30% (6/20)–IG 38,2% (13/34); *p*-value 0.541 4 months CG: 67.7% (21/31) – IG: 48.8% (21/43) vs. CG 32.3% (10/31) vs. IG: 51.2% (22/43) *p*-value 0.105 8 months GC 59.5% (22/37)–IG: 52.8% (28/53) vs. CG 40.5% (15/37)–IG 47.2% (25/53) *p*-value 0.53	
**Koczy et al. (2011) [32]** cluster RCT	*Y*	100% not restrained (free) 3 months CG 8.8% vs. IG 16.8% OR 2.16 (IC 95% 1.05–4.46) Restraint 3 months CG: 114/125 (91.2%); IG: 173/208 (83.2%)	Falls 3 months GI 16.3% vs. GC 8.0%; OR 2.08 (IC 95% 0.98–4.40)
**Köpke et al. (2012) [15]** RCT	*Y*	Any physical restraint 3 months CG 30.5 (26.6–34.4) vs. IG 23.9 (19.3–28.5) MD 6.6%; 95% CI (0.6–12.6) Cluster adjusted OR 0.72; 95% CI (0.53–0.97) *p*-value 0.03; ICCC 0.029 6 months Difference 6.5%; 95% CI (0.6–12.4) Cluster adjusted OR 0.71; 95% CI (0.52–0.97) *p*-value 0.03; ICCC 0.029	Residents ≥1 fall during period study Difference 3%; 95% CI (−3.5 to 9.4) Cluster adjustice OR 0.85; 95% CI (0.60 to 1.21) Fractures during period study Difference 0.5%; 95% CI (−0.5 to 1.4) OR (95% CI) = 0.76 (0.42 to 1.38)
**Pellfolk et al. (2010) [9]** cluster RCT	*Y*	Physical restraint 6 months CG 38.1% (53/139); IG 20.1% (30/149) *p*-Value baseline/ 6 months: 0.78/0.001 OR 0.21, 95% CI 0.08–0.57, *p*-value 0.002 restrained baseline vs. unrestraint 6 months CG 3.6% (n = 1/28) vs. IG 31.3% (n = 10/32) (*p* = 0.007). unrestrained baseline vs. restrained 6 months CG 23.4% (n26/111) vs. IG 6.8% (n8/117) (*p*-value 0.001).	Falls 6 months IG 10.1% vs. CG 8.6% *p*-Value baseline/ Follow-Up: 0.45/0.68
**Rovner et al. (1996) [35]** RCT	*Y*	Physical restraint 6 months CG 20/38 (52.6%) vs. IG 14/41 (34.1%) OR 0.47 [95% CI 0.19 to 1.16] *p*-value = 0.10	
**Testad et al. (2005) [33]** RCT	*Y*	Frequency of use of restraint–mean (range) 7 months CG 4/55; IG 2/96 CG 3.7 (0–25); GI 1.5 (0–10); *p*-value = 0.016	
**Testad et al. (2010) [34]** RCT	*Y*	Structural restraint 6 months CG 23/70 (33%); IG 48/75 (64%) 12 months CG 6/70 (13%) IG 8/75 (18%)	
**Testad et al. (2016) [26]** cluster RCT	*Y*	Change in any physical restraint physical restraint prevalence from baseline to /7months CG t0 10.5% vs. t1 6.1% *p* < 0.001; IG t0 14.5% vs. t1 10.5% *p*-value 0.007	

* MA, meta-analysis; *Y*, yes: studies included in meta-analysis; *N*, Not included; CG, control group; IC, intervention group; RE, educational rehabilitation; REC, educational rehabilitation with consultation.

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
