# Peer review of "The Effectiveness of Educational Training or Multicomponent Programs to Prevent the Use of Physical Restraints in Nursing Home Settings: A Systematic Review and Meta-Analysis of Experimental Studies"

_ijerph, 2020, doi:10.3390/ijerph17186738_

Round 1
Reviewer 1 Report
Dear Authors:
Congratulations for your job.
I have some suggestion:
- Aims was define to assess the effectiveness of an educational training or multicomponent program ti prevent the use of PR in nursing home. The secondary outcome was defined as decrease in rate of patient falls and injuries related to falls. I suggest to explain better this secondary outcome; I thought that you are try to get information about the change (increase or decrease) of falls relatives with the changes in PR as a result of the interventions. In design paragraph, you describe in line 102 another secondary outcomes that can confuse to readers. The same thing happen on line 238.
- Improve the tables, especially the table 2. There is a lot information and I think that you can find a better way to summary and show the most important.
- After reading his work, I think that it is very complex to analyze different educational trainings as if they were similar. You mention differences in the definitions used, in the methodology used for data collection, variability with respect to the samples, in addition to the differences in the interventions. Considering that the Cochrane review was published in 2011, and that your study searches for articles from 1996, I suggest making explicit which articles are shared with the Cochrane review and which are the new ones that would tip the balance in a different conclusion than what has been published-
Kind regards,
Reviewer 2 Report
This is a subject very close to my heart and professional interests and I thank you for carrying out this review. AS far as i can see, the review is meticulously conducted and comprehensively described and the outcomes are very important one for nurses and others involved in the care of older people - especially those with dementia. The standards regarding restraint - and not only for risk of injury, but also for administering nutrition in some parts of the world vary. Also, some areas of the world are more risk averse than others. Yet, physical restraint is almost always harmful to the recipient and should be reduced to a minimum or eradicated completely. For those involved in this endeavour, this is an excellent reference point. The study is Prospero registered and folllows PRISMA and Cochrane principles where appropriate. I look forward to seeing this one in print.
Author Response
Dear Reviewer,
thank you for your very positive feedback.
Reviewer 3 Report
Dear Authors,
Thank you for your study, the topic is important for nursing clinical practice.
I found your study interesting, however there are some aspects which should be clarified and corrected:
- manuscript presents huge material and therefore, it is difficult to read it in its current form. Especially the results part - too much of information and details given in too many tables. I am aware that the data is interesting and important, but you should re-consider what should be included in this piece of work and what left for another one.
Other issues:
- in the introduction: there are some awkward sentences, like e.g.: "However, evidence stress that PR use is not an adequate approach for reducing and preventing falls and injuries related falls because leads to an increase in risks" - in risk of what?
- Aim: it is unclear to me what have you meant by this sentence: "The secondary outcome is the decrease in rate of patient falls and injuries related to falls". I am not sure it can be an aim of the study.
- Methods - when describing searching strategy, there are some repetitions, e.g. the types of sources are actually given two times (at the beginning, when describing databases and then when giving inclusion criteria), please re-write to avoid repetition
- conclusions should be re-written - in this current form it is more the discussion than conclusions.
Reviewer 4 Report
It seems to me a very interesting manuscript that should be published. Congratulations to the authors for the work done.
As suggestions, I would like to tell authors to describe what they are educational training or multicomponent programs and what previous experiences exist in other countries in the introduction section.
Author Response
Dear Reviewer,
thank you for your positive feedback.
Round 2
Reviewer 3 Report
Dear Authors,
Thank you for providing suggested changes.
I have one technical issue: Table 2 - there is 3rd column which is empty - it should be deleted.